Assisted migration and the rare endemic plant species: the case of two endangered Mexican spruces

Mendoza-Maya Eduardo 1
Gómez-Pineda Erika 2
Sáenz-Romero Cuauhtémoc 3
http://orcid.org/0000-0002-3284-422X Hernández-Díaz José Ciro 4
http://orcid.org/0000-0002-5135-5739 López-Sánchez Carlos A. 5
http://orcid.org/0000-0001-7422-4953 Vargas-Hernández J. Jesús 6
http://orcid.org/0000-0002-2954-535X Prieto-Ruíz José Ángel 7
http://orcid.org/0000-0002-2341-5458 Wehenkel Christian 4 wehenkel@ujed.mx
1 Programa Institucional de Doctorado en Ciencias Agropecuarias y Forestales, Universidad Juárez del Estado de Durango , Durango , México
2 Centro de Investigaciones en Geografía Ambiental, Universidad Nacional Autónoma de México , Morelia, Michoacán , México
3 Instituto de Investigaciones sobre los Recursos Naturales, Universidad Michoacana de San Nicolás de Hidalgo , Morelia, Michoacán , México
4 Instituto de Silvicultura e Industria de la Madera, Universidad Juárez del Estado de Durango , Durango, Durango , México
5 SMartForest Group, Department of Biology of Organisms and Systems, Mieres Polytechnic School, Universidad de Oviedo , Mieres , Spain
6 Postgrado en Ciencias Forestales, Colegio de Postgraduados , Montecillo, Texcoco, Edo. de México , México
7 Facultad de Ciencias Forestales y Ambientales, Universidad Juárez del Estado de Durango , Durango, Durango , México
Fenu Giuseppe
Electronic publication date: 2022 Aug 3
Publication date: 2022
Volume: 10
Electronic Location ID: e13812
Received 2022 Mar 9; Accepted 2022 Jul 7
Copyright: © 2022 Mendoza-Maya et al.
Copyright year: 2022
Copyright holder: Mendoza-Maya et al.
License: This is an open access article distributed under the terms of the Creative Commons Attribution License, which permits unrestricted use, distribution, reproduction and adaptation in any medium and for any purpose provided that it is properly attributed. For attribution, the original author(s), title, publication source (PeerJ) and either DOI or URL of the article must be cited.
License URL: https://creativecommons.org/licenses/by/4.0/

Keywords: Conservation of threatened species, Climate change, Narrow endemics, Paleodistributions, Picea martinezii, Picea mexicana, Species distribution modelling

Funding: Forest Genetic Resources Working Group Mexican Council of Science and Technology and the Mexican National Forest Commission Joint Fund CONACyT-CONAFOR-2017-4-292615 Eduardo Mendoza-Maya 349164 Erika Gómez-Pineda 339216 The study reported in this paper is an undertaking of the Forest Genetic Resources Working Group, of the North American Forestry Commission. Funding was provided by the Mexican Council of Science and Technology and the Mexican National Forest Commission Joint Fund (CONACyT-CONAFOR-2017-4-292615) to Christian Wehenkel. CONACyT granted graduate fellowships to Eduardo Mendoza-Maya (349164) and Erika Gómez-Pineda (339216). There was no additional external funding received for this study. The funders had no role in study design, data collection and analysis, decision to publish, or preparation of the manuscript.

==============================
Background

In the projected climate change scenarios, assisted migration might play an important role in the ex situ conservation of the threatened plant species, by translocate them to similar suitable habitats outside their native distributions. However, it is unclear if such habitats will be available for the Rare Endemic Plant Species (REPS), because of their very restricted habitats. The aims of this study were to perform a population size assessment for the REPS Picea martinezii Patterson and Picea mexicana Martínez, and to evaluate the potential species distributions and their possibilities for assisted migration inside México and worldwide.

Methods

We performed demographic censuses, field surveys in search for new stands, and developed distribution models for Last Glacial Maximum (22,000 years ago), Middle Holocene (6,000 years ago), current (1961–1990) and future (2050 and 2070) periods, for the whole Mexican territory (considering climatic, soil, geologic and topographic variables) and for all global land areas (based only on climate).

Results

Our censuses showed populations of 89,266 and 39,059 individuals for P. martinezii and P. mexicana, respectively, including known populations and new stands. Projections for México indicated somewhat larger suitable areas in the past, now restricted to the known populations and new stands, where they will disappear by 2050 in a pessimistic climatic scenario, and scarce marginal areas (p = 0.5–0.79) remaining only for P. martinezii by 2070. Worldwide projections (based only on climate variables) revealed few marginal areas in 2050 only in México for P. martinezii, and several large areas (p ≥ 0.5) for P. mexicana around the world (all outside México), especially on the Himalayas in India and the Chungyang mountains in Taiwan with highly suitable (p ≥ 0.8) climate habitats in current and future (2050) conditions. However, those suitable areas are currently inhabited by other endemic spruces: Picea smithiana (Wall.) Boiss and Picea morrisonicola Hayata, respectively.

Conclusions

Assisted migration would only be an option for P. martinezii on scarce marginal sites in México, and the possibilities for P. mexicana would be continental and transcontinental translocations. This rises two possible issues for future ex situ conservation programs: the first is related to whether or not consider assisted migration to marginal sites which do not cover the main habitat requirements for the species; the second is related to which species (the local or the foreign) should be prioritized for conservation when suitable habitat is found elsewhere but is inhabited by other endemic species. This highlights the necessity to discuss new policies, guidelines and mechanisms of international cooperation to deal with the expected high species extinction rates, linked to projected climate change.

Introduction

To know the species distributions is fundamental for their conservation. Certainly, understanding the species’ niche requirements and habitat specificity is essential to define the possibilities for management in the context of climate change. As linked ecological properties, niche requirements and habitat specificity influence geographical ranges of taxa (Crain et al., 2015) and ultimately, it determines commonness, endemism and rarity. This last ecological property, shared by 36.5% of the global plant diversity (Enquist et al., 2019), needs special attention because involves taxa with strong influence on ecosystem services (Mouillot et al., 2013) and high vulnerability to extinction (Işik, 2011).

Assisted migration is an ex situ conservation approach that emerged as a response to the imminent decoupling between climate and species in natural reserves (Peters & Darling, 1985). Pedlar et al. (2012) distinguished two types: forestry assisted migration, the objective of which is to maintain forest health and productivity; and species rescue assisted migration, whose goal is to avoid extinctions of threatened species.

Rescue assisted migration, the approach considered in this study (hereafter named only assisted migration), has been recognized as a viable method to conserve vulnerable species, by translocating them to similar suitable habitats outside their native ranges, where they can reproduce and compete successfully (McLane & Aitken, 2012). However, this adaptation strategy continues to be debated, pointing to the potential risk that translocated species could become invasive or could serve as vectors of new pests and diseases (Schwartz et al., 2012; Simler et al., 2019; Butt et al., 2021).

Additionally, if assisted migration becomes a regular ex situ conservation strategy, some questions remain unanswered regarding the Rare Endemic Plant Species (REPS): (i) Is it possible to find potential habitats outside the natural range of REPS, considering their high habitat specificity? (ii) If new suitable environments are found, is the area large enough to establish populations of a minimum viable size, able to survive in the long-term? (iii) Furthermore, what can be done if a new suitable habitat is found elsewhere, but is already occupied by other REPS? We do not have the answers to these questions, mainly because field studies of assisted migration with REPS are scarce (Butt et al., 2021).

Species Distribution Modelling (SDM), based on empirical associations between species occurrences and environmental variables, has become an important tool to understand current species distributions and for designing management and conservation strategies (López-Tirado & Hidalgo, 2016; Bosso et al., 2017; Ongaro et al., 2018; Sofaer et al., 2019), including assisted migration (McLane & Aitken, 2012). Regarding the rare endemic species, SDM has been improved by a variety of algorithms and methodologies for model construction, in addition to the incorporation of environmental factors with different scales of influence on species distributions (e.g. Pearson et al., 2007; Williams et al., 2009; Patsiou et al., 2014; McCune, 2016; Mi et al., 2017; Feng et al., 2022).

Previous studies have proposed systematic decision-making guides for assisted migration (Pérez et al., 2012), have evaluated the need and potential for assisted migration in different taxa through the SDM approach (Hällfors et al., 2016) and have used SDM to delimit current distributions of rare species at local or regional scales (e.g. McCune, 2016), or to evaluate future impacts of climate change, mainly by using climatic variables (Ledig et al., 2010; Pinedo-Alvarez et al., 2019). Nevertheless, studies on the potential value of assisted migration for REPS, considering the most complete sets of environmental variables of recognized influence on species distributions (e.g. Penteriani et al., 2019; Barrio-Anta et al., 2020; López-Sánchez et al., 2021), or global habitat searches using climatic variables, are scarce.

In this study we explore the potential of assisted migration as a tool for the conservation of two Mexican REPS (both at the category of endangered; IUCN, 2021): Picea martinezii Patterson and Picea mexicana Martínez. These species are relicts of the last glacial age, confined to very specific habitats, with scattered, fragmented and few isolated populations (Ledig et al., 2000a). In addition to other threats distinctive of the rare endemic species (Işik, 2011; Cogoni et al., 2019), such as their low genetic diversity (Ledig et al., 2000b; Ledig, Hodgskiss & Jacob-Cervantes, 2002) and decreased reproductive fitness (Flores-López, López-Upton & Vargas-Hernández, 2005; Flores-López et al., 2012), these spruces are threatened by climate change, as indicated by projected alterations in temperature and rain regimes on Mexican temperate forests (Sáenz-Romero et al., 2010), particularly in sites where these species thrive (Ledig et al., 2010), which could increase tree mortality through hotter-drought events (Hammond et al., 2022).

Since the discovery of P. mexicana (Martínez, 1961) and P. martinezii (Müller-Using & Alanis, 1984; Patterson, 1988), a total of seven populations had been roughly documented, but the population sizes, the exact extent and spatial distribution of these populations remain relatively unknown, and exploration to discover new stands had not been completed. Both spruces represent proper models for testing the viability of assisted migration in the ex situ conservation of rare species with restricted habitat.

We focused our study on three aspects of these REPS: (i) the exploration of potential new stands and performance of a population size assessment as a starting point for future demographic evaluations; (ii) construction of models describing the potential past and current distributions of these species; and (iii) identification of areas outside their current ranges (inside México and worldwide) with probability of harboring suitable habitats for future assisted migration, by modelling future distributions. Our hypothesis is that the suitable climatic habitat of P. martinezii and P. mexicana can be found elsewhere outside their current ranges and hence, assisted migration is a viable tool for the ex situ conservation of both species, considering future projections of climatic change.

Materials and Methods

Study area, exploration of new stands and species distribution data

The study area is located in the Sierra Madre Oriental (SMOr) and Sierra Madre Occidental (SMOc), two parallel mountain ranges that cross northern México from north to south; SMOr alongside the Gulf of México, SMOc alongside the Pacific Ocean. Both mountain ranges are connected east-west by the Trans-Mexican Neovolcanic Axis (TMNVA) in central México (Fig. 1). Picea martinezii is only located in the northern SMOr, in four populations: El Butano, Agua de Alardín, Agua Fría and La Encantada, state of Nuevo León (Table 1, Fig. 1). These populations are generally found on north-facing slopes, near creeks, ravines or cliffs in the montane cloud forests at elevations ranging from 1,800 to 2,500 m (Ledig et al., 2000a). There are only three documented populations of Picea mexicana on the north-facing slopes of the highest peaks of the northern SMOc (one single population: El Mohinora, state of Chihuahua) and on SMOr (two populations: La Marta and El Coahuilón, state of Coahuila), in the conifer forests of the subalpine zones in elevations ranging from 3,000 to 3,600 m (Table 1, Fig. 1) (Ledig et al., 2000a). Detailed descriptions of the Mexican montane cloud and subalpine forest vegetation types are provided by Rzedowski (2006).

Figure 1 Locations of the four known contemporary populations of Picea martinezii (circle symbols) and the three known populations of Picea mexicana (star symbols).

Prominent geographic regions of the Sierra Madre Occidental (SMOc, dark blue), Sierra Madre Oriental (SMOr, bluish green) and Trans-Mexican Neovolcanic Axis (TMNVA, vermilion) are shown. Note: The population of Agua de Alardín is also known as Agua Lardín. The SMOc, SMOr and TMNVA provinces file shapes were taken from Morrone, Escalante & Rodríguez-Tapia (2017), and the topographic base map for all figures was taken from ESRI (2018).

Table 1 Populations of Picea martinezii and Picea mexicana used to characterize the habitats and for modelling distributions of both species.

Population or
stand	Code	Municipality
and Statea	Elevation
(m)	Coordinates
(decimals)	Status (number
of individualsb)	Area
(ha)	
Picea martinezii	
El Butano	EB	Montemorelos, NL	2,180	25.178N
−100.126W	Known (1,253)	23.0	
						
Agua de Alardínc	AL	Aramberri, NL	2,120	24.042N
−99.734W	Known (84,498)	74.3	
						
Agua Fría	AF	Aramberri, NL	1,820	24.038N
−99.710W	Known (2,769)	53.8	
						
La Encantadad	LE	Zaragoza, NL	2,515	23.890N
−99.791W	Known (712)	5.2	
						
La Encantada 2	LE-2	Zaragoza, NL	2,378	23.890N
−99.778W	New (12)	0.1	
						
Zaragoza	ZA	Zaragoza, NL	2,483	23.890N
−99.773W	New (22)	1.0	
						
Total					(89,266)	157.4	
Picea mexicana							
El Mohinora	EM	Guadalupe y Calvo, Chih.	3,113	25.961N
−107.042W	Known (11,383)	33.0	
						
El Coahuilón	EC	Arteaga, Coah.	3,528	25.247N
−100.354W	Known (2,253)	49.0	
						
La Marta	LM	Arteaga, Coah.	3,494	25.198N
−100.364W	Known (17,728)	41.6	
						
La Marta 2	LM-2	Arteaga, Coah.	3,393	25.213N
−100.413W	New (6,300)	24.0	
						
La Marta 3	LM-3	Arteaga, Coah.	3,475	25.205N
−100.386W	New (35)	6.5	
						
La Marta 4	LM-4	Arteaga, Coah.	3,364	25.203N
−100.369W	New (50)	1.7	
						
La Marta 5	LM-5	Arteaga, Coah.	3,096	25.207N
−100.369W	New (60)	1.0	
						
El Mohinora 2	EM-2	Guadalupe y Calvo, Chih.	3,139	25.957N
−107.029W	New (1,250)	16.3	
						
Total					(39,059)	173.1	
Notes:

a NL = Nuevo León; Coah. = Coahuila (both in the Sierra Madre Oriental); Chih. = Chihuahua (in the Sierra Madre Occidental).

b Including trees, saplings and natural regeneration (heights > 30 cm).

c Agua de Alardín (also known as Agua Lardín) = Cañada El Puerto (I, II, III) in Ledig et al. (2000a).

d La Encantada = La Tinaja in Ledig et al. (2000a).

The distribution models were based on presence/absence data. In order to explore the potential existence of new stands and get more presence records for SDM, high-elevation, moist, cold, north exposure sites were surveyed in the surroundings of the known populations (at distances of up to 39 km). The potential sites were identified from maps of suitable climatic habitat for P. martinezii and P. mexicana, projected under contemporary climate (average 1961–1990) provided by Ledig et al. (2010), as well as from unconfirmed oral testimonies given by local foresters and landowners; this allowed us to get 46 to 50 presence records of P. martinezii and P. mexicana, respectively (Tables 1, S1, S2 and S3). All these presences were recorded in the center and periphery of the known populations and new stands, during the population size assessments that considered all individuals taller than 30 cm (including recruitment, saplings and trees), which were carried out as part of this study in 2018 and 2019 (Table 1). Absences records (22,004 to 32,571) were sampled from the sites listed in the Mexican National Forest and Soil Inventory (MexFI), developed by the Mexican National Forest Commission (CONAFOR, 2009) (Table S3). Model projections for mapping the distribution of suitable areas for both tree species were performed for the whole Mexican territory and for all the world land areas (excluding Antarctica).

Environmental variables and species distribution modelling

Forty two environmental variables of different classes: climate, topography, soil and geology (e.g. succesfully used by Penteriani et al., 2019; Barrio-Anta et al., 2020; López-Sánchez et al., 2021) were considered possible predictors of the distribution of P. martinezii and P. mexicana. Gridded data of 19 climatic variables (1961–1990 reference period) were retrieved at 30 arc-second resolution from WorldClim dataset (Hijmans et al., 2005, available at URL: https://worldclim.org); data of 14 soil variables were obtained from the SoilGrids250m at 250 m × 250 m resolution (available at URL: https://www.soilgrids.org), a repository of the spatial distribution of soil properties across the globe (Hengl et al., 2017). Data on two geological (30 arc-second resolution) and seven topographic variables (250 m × 250 m resolution) were obtained from digital models provided by the Mexican National Institute of Statistics, Geography and Informatics (INEGI) (available at URL: http://www.inegi.mx) (Table S4). Presences records were interleaved with each raster layer of the analyzed environmental variables, which were resampled at 30 arc-second cell resolution with the nearest neighbor method. Then, mean values of the environmental variables were extracted from pixels holding the presence records.

The varying methodologies for SDM may influence the final model metrics and projections, and the need to evaluate such methods in this kind of projects has previously been recommended (e.g. Pearson et al., 2006; Qiao, Soberón & Peterson, 2015). Regarding the rare species with narrow distributions, some methods have shown better results (Mi et al., 2017). Distribution models for P. martinezii and P. mexicana were constructed with the non-parametric regression Random Forest (RF) algorithm including cross validation, based on its higher performance than other methods (including MaxEnt) to predict the rare species distributions (Mi et al., 2017), and following a similar methodology as Penteriani et al. (2019), Castaño-Santamaría et al. (2019), and Barrio-Anta et al. (2020). In brief, RF constructs a set of regression and classification trees using different independent variables randomly selected from the complete data set (Breiman, 2001; Deschamps et al., 2012). To include only the main predictors shaping the species distributions (Hall, 1999), collinearity between variables was evaluated before model construction. This was performed with the open source WEKA software (Hall et al., 2009), using the wrapper methodology (Zhiwei & Xinghua, 2010) which selects the best ranked variables through the Variable Importance Measure (VIM) function (see the Results section and Table 2 for the list of selected variables included in final models). Variables of different scales were normalized following the methodology of Castaño-Santamaría et al. (2019) to make them comparable, and VIM values were expressed adding up to a unitary value (normalized importance), which can also be expressed in percentage (Table 2). Spatially continuous maps were generated by applying the final models to environmental spatial variables resampled to a 30 arc-second resolution.

Table 2 Environmental variables of greatest importance in the Picea martinezii and Picea mexicana distribution models for México and the world at 30 arc-second, as indicated by the percentage of normalized importance of the Variable Importance Measure (VIM) function.

Class	Variable	Description	Normalized importance (%)	
Picea martinezii model for México	
Topography	PRC	Profile curvature	41.89	
Soil	SC	Soil organic carbon content (g kg−1)	32.46	
Climate	Bio_01	Annual mean temperature (°C)	25.65	
Picea martinezii global model	
Climate	Bio_02	Mean diurnal range (mean of monthly (max temp − min temp)) (°C)	22.90	
Climate	Bio_13	Precipitation of wettest month (mm)	17.85	
Climate	Bio_04	Temperature seasonality (standard deviation * 100) (°C)	17.57	
Climate	Bio_07	Temperature annual range (BIO5–BIO6) (°C)	17.48	
Climate	Bio_08	Mean temperature of wettest quarter (°C)	15.28	
Climate	Bio_19	Precipitation of coldest quarter (mm)	8.93	
Picea mexicana model for México	
Climate	Bio_06	Minimum temperature of coldest month (°C)	18.27	
Topography	WI	Wetness index	13.89	
Soil	SC	Soil organic carbon content (g kg−1)	13.75	
Soil	BD	Bulk density of the fine earth fraction (<2 mm) (kg m−3)	12.71	
Topography	ASP	Aspect (°)	11.89	
Climate	Bio_09	Mean temperature of driest quarter (°C)	11.77	
Climate	Bio_17	Precipitation of driest quarter (mm)	9.83	
Geology	Geo	Geological units	7.90	
Picea mexicana global model	
Climate	Bio_04	Temperature seasonality (standard deviation * 100) (°C)	17.72	
Climate	Bio_16	Precipitation of wettest quarter (mm)	15.73	
Climate	Bio_08	Mean temperature of wettest quarter (°C)	14.49	
Climate	Bio_11	Mean temperature of coldest quarter (°C)	14.09	
Climate	Bio_18	Precipitation of warmest quarter (mm)	13.59	
Climate	Bio_01	Annual mean temperature (°C)	13.57	
Climate	Bio_14	Precipitation of driest month (mm)	10.82	

We used the k-fold cross validation approach (k-fold = 10) to test the precision (repeated 10 times) of the RF classifier on unseen data. This was done by dividing the data set into k subsets and using one subset as the test set and the other k-1 subsets as the training set, each time the model was applied. The accuracy of the model predictions was evaluated with the confusion matrix that shows the four-way classification of a sampled point. From this last evaluation, we calculated the following model metrics, widely used in SDM studies (Freeman & Moisen, 2008; Penteriani et al., 2019): (i) the Area Under the Receiver Operating Characteristic Curve (AUC); (ii) the Overall Accuracy (OA); (iii) Matthews Correlation Coefficient (MCC); (iv) the True Skill Statistic (TSS), (v) Cohen’s Kappa, (vi) Sensitivity; and (vii) Specificity.

To map the species distributions, thresholds for the probability of presence (PoPthreshold) were selected for each species by combining two approaches: (i) the method that minimizes the difference between the absolute values of sensitivity and specificity (Jiménez-Valverde & Lobo, 2007); and (ii) the method that requires an appropriate fixed specificity (Freeman & Moisen, 2008), in this case, based on proper probability values around the thresholds obtained with the first approach (see the Results section and Table S5). The last approach mentioned has been recommended particularly for rare species when it is important to include all possible populations in planning. Both approaches for threshold selection were based on the evaluation of models constructed for current conditions for México (see next section) and the real species presences/absences (Table S5). The final PoPthreshold values were used to map the two species distributions in all projections in the hyperspace.

Past, contemporary and future distributions: projections for México

The fitted models were projected onto spatial projections of the most important environmental variables (Table 2) at a 30 arc-second resolution, for the current conditions (1961–1990 reference period) to estimate the contemporary potential distributions of these species. Additionally, the following projections were performed using the Community Atmospheric Model scenario version 4 (CCSM4): (1) to the paleoclimate data of the Last Glacial Maximum (LGM, ~22,000 thousand years ago = 22 ka) and the Middle Holocene (MH, ~6 ka); and (2) to the future periods centered on years 2050 and 2070 under two different Representative Concentration Pathways (RCPs) (IPCC, 2013): a moderate scenario (RCP 4.5), which assumes a total radiative forcing stabilized at 4.5 Wm2 by 2100; and, a pessimistic scenario (RCP 8.5) which considers a higher radiative forcing of 8.5 Wm2 by 2100. Data sets for current climatic conditions were obtained from WorldClim version 1.4 (Hijmans et al., 2005, available at URL: http://www.worldclim.com) and data of the CCSM4 from the National Center for Atmospheric Research (available at URL: https://www.cesm.ucar.edu/models/ccsm4.0/). The equivalence in surface area considered for each projected pixel of suitable habitat was 0.7 km2 for latitudes corresponding to the Mexican territory (between 15° to 31° LN).

Contemporary and future distributions: global projections

We also estimated the potential contemporary (1961–1990 reference period) and future (period centered in 2050) distributions of the suitable climate habitats at a global scale, following the previously described methodology for model construction, but considering only the 19 climatic variables available in WorldClim (Tables S4 and 2). The resolution of the climatic data was 30 arc-second, which is the highest uniform resolution available for the entire world. Projections to the future period were performed using the CCSM4. We intended global cautionary projections for the two studied tree species, by considering the most pessimistic climatic scenario (RCP8.5) in an intermediate future period. All distribution maps were created in QGIS v.3.16.1 (QGIS Development Team, 2020).

Results

Demographic census, area extent and exploration of new populations

Population size assessments showed 89,266 P. martinezii individuals (including recruitment, saplings and trees) distributed in a total area of 157.4 ha, and 39,059 P. mexicana individuals in a total area of 173.1 ha (Table 1). The largest P. martinezii population was Agua de Alardín with 84,498 individuals covering an area of 74.3 ha, and the smallest was La Encantada with 712 individuals and 5.2 ha. The largest population of P. mexicana was La Marta, with 17,728 individuals covering an area of 41.6 ha; the smallest was El Coahuilón, with 2,253 individuals scattered in an area of 49 ha (Table 1).

On the other hand, seven new (previously unreported) stands were discovered and explored, with numbers of individuals per stand ranging from 12 to 6,300 and areas between 0.1 and 24 ha. All new stands were located close (0.5 to 5.0 km) and at similar elevations to the previously known populations in both the Sierra Madre Oriental (SMOr) and Sierra Madre Occidental (SMOc) (Table 1).

Species distribution modelling and model assessment

The normalized importance scores of the VIM function, selected three variables as the most important predictors for P. martinezii, and eight variables for P. mexicana in the models for México (Table 2). In the global models, the same importance analysis selected six and seven climatic variables for P. martinezii and P. mexicana, respectively (Table 2).

According to the model metrics AUC, OA, MCC, TSS, Kappa, Sensitivity and Specificity with values ≥0.9, the t goodness-of-fit were highly accurate (Table S3). The PoPthreshold values for P. martinezii and P. mexicana were 0.73 and 0.83, respectively, based on the sensitivity-specificity balance approach. With a probability value of 0.8 or more, the correct presence prediction was 100% for P. martinezii and more than 55% for P. mexicana (Table S5). Hence, based on the fixed specificity approach, all the performed projections in the hyperspace considered a PoPthreshold of 0.8 to denote the presence (above this value) or absence (under the value) of the highly suitable habitat. However, to show the area holding less suitable habitat, a minimum probability of presence of 0.5 was considered too. Finally, two categories of presence were used for displaying the results: 0.5–0.79 = intermediate and 0.8–1.0 = high.

Mapping suitable habitat: projections for México and the World

Projections of suitable habitat during the Last Glacial Maximum (LGM) and the Middle Holocene (MH) indicate overall a very small and fragmented distribution for both species. In particular for P. martinezii, it was projected only 15.4 km2 (probability ≥ 0.5) during the LGM, and 12.6 km2 (none of them with p ≥ 0.8) for the MH (Fig. 2); all those pixels were found highly scattered, mainly at the Trans-Mexican Neovolcanic Axis (Central México) and at the Sierra Madre Oriental, close to the contemporary distribution (near the border of Nuevo León and Tamaulipas states) (Figs. 3A and 3B). Picea mexicana had a maximum of 423.5 km2 of suitable habitat (p ≥ 0.5) during the LGM (Fig. 2), mostly at the Trans-Mexican Neovolcanic Axis, with an important area at northwest of Veracruz state, Sierra Madre Oriental (Fig. 3C, a, b, c), an area where the species is completely absent today. It is interesting to notice that suitable habitat of P. mexicana was absent at the Sierra Madre Occidental during the LGM (Fig. 3C), and then was present during the MH (Fig. 3D, a-d), around the place where the contemporary population of El Mohinora is located (Fig. 3D, c).

Figure 2 Projected area of suitable habitat for P. martinezii and P. mexicana in México and the world at different times.

Projections for México, made after modeling considering climate, topographic, soil and geological variables, included the Last Glacial Maximum (LGM, ~22 thousand years ago = ka), Middle Holocene (MH, ~6 ka), current (1961–1990 period), and future time (decades 2050 and 2070, both with RCP 4.5 and RCP 8.5). Worldwide projections, made after modeling using only climate variables, included the 1961–1990 period (Wcurrent) and the decade 2050–2060 with RCP 8.5 (W2050 RCP 8.5). Pixel counts for global projections excluded predicted areas for México in this Figure. All periods show projections made at 30-arc second resolution and pixels with probabilities of occurrence of 0.5–0.79 (yellow) and 0.8–1.0 (vermilion). Numbers above the bars indicate the projected number of pixels.

Figure 3 Potential distribution of the habitat for Picea martinezii and P. mexicana in the Last Glacial Maximum (~22 thousand years ago = ka) (A and C) and the Middle Holocene (~6 ka) (B and D).

The Last Glacial Maximum projection for P. martinezii shows the less suitable habitat on the Mexican territory (A-a to l), with only one pixel holding the highly suitable habitat in the northwest of Veracruz state (A-h); the Middle Holocene projection shows the distribution of the less suitable areas (B-a to i). For P. mexicana, the Last Glacial Maximum projection shows suitable areas in the center of México at northeast of Hidalgo state (C-a), borders of Puebla and Hidalgo states (C-b), and west of Veracruz state (C-c), with some less suitable areas in Estado de México (C-d and e) and Chiapas states (C-f); the Middle Holocene projection shows suitable areas in northern México, at north (D-a) and center (D-b) of Chihuahua, borders of Chihuahua and Durango states (D-c), south of Durango (D-d), borders of Coahuila and Nuevo León states (D-e), and south of Nuevo León state (D-f).

Contemporary projections (reference period 1961–1990) of highly suitable habitat distribution predicted perfectly all the actual populations for both species (Figs. 4A and 4B, a, b), which represent an area of 4.9 km2 for P. martinezii and 13.3 km2 for P. mexicana (Fig. 2). Besides, the predicted highly suitable areas outside the current distribution were totally absent for both species. For P. martinezii, there were very few scattered isolated pixels (all with probability 0.5 to 0.79) at northwest of Chihuahua state (Fig. 4A, c), along the TMNVA (Fig. 4A, d, e, f), and at the Sierra Madre del Sur (states of Guerrero, Oaxaca and Chiapas; Fig. 4A, g, h, i). For P. mexicana, there were even less regions with predicted pixels outside its natural distribution in the SMOc and SMOr (all of them with probabilities 0.5 to 0.79; Fig. 4B, c, d).

Figure 4 Actual and potential distribution of the suitable habitat in current conditions (1961–1990 reference period) for P. martinezii (A) and P. mexicana (B).

Projection for P. martinezii shows the highly suitable habitat (probability ≥ 0.8) restricted to the natural populations (EB = El Butano, AL = Agua de Alardín, AF = Agua Fría, and LE = La Encantada) in the center west (A-a) and southeast (A-b) of Nuevo León state, and less suitable areas (probability 0.5 to 0.79) spread in northern (A-c) and central (A-d to A-i) México. Projection for P. mexicana shows the presence of the highly suitable habitat restricted to the natural populations (EM = El Mohinora, EC = El Coahuilón, and LM = La Marta) (B-a and B-b), and less suitable areas near the populations at center (B-c) and south (B-d) of Nuevo León state.

Projections of suitable habitat to the future within México, under climatic change scenarios, indicate a severe reduction for both species’ habitats (Figs. 2 and 5). Suitable habitat (p ≥ 0.5) predicted for P. martinezii drops from a total of 15.4 km2 at present, to only 7.0 and 2.8 km2 for 2050, scenarios RCP 4.5 and 8.5, respectively. For year 2070, the drop is even further, to have just 3.5 and 2.1 km2 for scenarios RCP 4.5 and 8.5, respectively (Fig. 2). And still worse: by 2050, the highly suitable habitat (p ≥ 0.8) disappears completely at and nearby all the current contemporary P. martinezii populations (Fig. 5A). For P. mexicana, projections show an even worse situation, with 0.7 km2 (a single pixel) for year 2050 RCP 8.5 and 1.4 km2 for year 2070 RCP 4.5; there is not a single pixel predicted for 2070 RCP 8.5 inside México (Figs. 2, 5C and 5D).

Figure 5 Potential distribution of the habitat (modeling based on climate, topography, soil and geological variables) for Picea martinezii (A-B) and P. mexicana (C-D) for 2050 and 2070, in both the moderate (RCP 4.5) and pessimistic (RCP 8.5) scenarios.

Blue pixels shared by RCP 4.5 and 8.5, mean that the same pixel is projected as suitable in both scenarios. The 2050 (A) and 2070 (B) projections for P. martinezii show the spread distribution of the less suitable habitat in the Mexican territory, with more area in the moderate (A-a to A-f in 2050; B-a to B-d in 2070) than in the pessimistic (A-a to A-d in 2050; B-a to B-c in 2070) scenarios, and only three pixels of the less suitable habitat remaining by 2070 in the pessimistic scenario at the west of Chihuahua (B-a), center of Estado de México (B-b) and Estado de México-Puebla border (B-c); these three sites are present in all projections for P. martinezii from current to future conditions. The 2050 (C) and 2070 (D) projections for P. mexicana show the highly suitable habitat only in the moderate scenario, on and near the current populations of La Marta and El Coahuilón in the Coahuila-Nuevo León border (C-a and D) and center of Nuevo León (C-b); in the pessimistic scenario, only one pixel with the less suitable habitat in 2050 (C-b) and no area at all (highly nor less suitable habitat) in 2070 (D).

Worldwide projections for P. martinezii, show similar and extremely grim scenarios, either for contemporary climate and for 2050 RCP 8.5: there is not a single pixel available outside México, but some areas in the Mexican territory, few of which will remain in the future (Figs. 2 and 6).

Figure 6 Global projections of the potential distribution of the climatically suitable habitat (based only on climatic variables) for Picea martinezii in current conditions (1961-1990 reference period) and for 2050 (pessimistic scenario).

All sites for the contemporary and future projections are located in México, on and around the current natural populations of El Butano (EB) at the Coahuila-Nuevo León border (a), the center of Nuevo León state (b), and at Agua de Alardín (AL), Agua Fría (AF) and La Encantada (LE) which are at southern Nuevo León state, where a reduced area holding the less suitable habitat will remain by 2050 (c, blue pixels).

Worldwide projections for P. mexicana show a quite different and complex picture outside México: a total of about 141,000 pixels appears suitable for contemporary climate, and nearly 119,000 pixels for year 2050 RCP 8.5 (Figs. 2 and 7). Thus, despite the drastic contraction or even vanishing of suitable area for P. mexicana in México (when modeling considered climate, soil, geology and topography; Figs. 5C and 5D), worldwide projections (considering only climate) indicate that there is and there will be suitable climatic habitat (p ≥ 0.5) in: Laurentian mountains in Canada, Appalachian Mountains in USA, the Andes in south Chile, the Pyrenees in the Spain-France border, the Alps in the France-Switzerland border, the Caucasus mountains at Georgia-Russia border, Khingan mountains in China, Sayan mountains in the Mongolia-Russia border, The Himalayans in India, Sobaek and Taebaek mountains in South Korea, Taebaek mountains in North Korea, the Japanese Alps in Japan, Chungyang mountains in Taiwan, and the Southern Alps in New Zealand (Fig. 7). For the projected year 2050 RCP 8.5, a large proportion of those suitable areas under contemporary climate will remain for P. mexicana in all regions, except in México, the Appalachian Mountains in USA and the Pyrenees in the Spain-France border (Figs. 2 and 7B). Outside México, the highly suitable habitat (p ≥ 0.8) for P. mexicana is currently available only in The Himalayans in India (Fig. 7A, j) and the Chungyang mountains in Taiwan (Fig. 7A, l), where 10 pixels (about 6.4 km2; Fig. 7B, g) and 25 pixels (about 18.0 km2; Fig. 7B, i), respectively, will remain by 2050.

Figure 7 Global projections of the potential distribution of the climatically suitable habitat for Picea mexicana in contemporary conditions (A) and 2050 (B).

All sites for the contemporary projection are located in many mountainous regions of the world, such as: Sierra Madre Occidental (A-a), Sierra Madre Oriental (A-b), Laurentian mountains (A-c), Appalachian Mountains (A-d), southern Andes (A-e), The Pyrenees (not shown), The Alps (A-f), the Caucasus mountains (A-g), Khingan mountains (A-h), Sayan mountains (A-i), The Himalayans (A-j), Sobaek-Taebaek mountains (A-k), the Japanese Alps (A-k), Chungyang mountains (A-l) and the Southern Alps (A-m), where the habitat will remain by 2050, except for México, the Appalachian Mountains and the Pyrenees. The highly suitable habitat by 2050 outside México coincide with the geographical distribution of other endemic spruce species: Picea smithiana (blue circles) on The Himalayans in India (A-j and B-g) and P. morrisonicola (yellow circles) on Chungyan mountains in Taiwan (A-l and B-i). Contemporary projection (A) shows the natural populations of El Mohinora (EM), La Marta (LM) and El Coahuilón (EC). Occurrences of P. smithiana and P. morrisonicola were obtained from the Global Biodiversity Information Facility (GBIF, 2021).

Discussion

Demographic census, area extent and exploration of new populations

The results of the first complete population size assessments, as well as better knowledge of population areas and delimitations, provide the basis for the monitoring and management of the two studied species, and therefore, are important for conservationists, local communities, stakeholders and governmental institutions. The seven new natural stands located by field surveys, showed that these species thrive only on very specific habitats at similar elevations than the previously known populations, both in the Sierra Madre Oriental and Sierra Madre Occidental. These new stands may be portions of the nearest known populations, as suggested by their proximity of 0.5 to 5.0 km to each other. Therefore, the total number of populations can be considered as previously reported for these spruce species: four populations of P. martinezii and three populations of P. mexicana (Ledig et al., 2000a). However, the new stand located at 5.0 km from La Marta, and holding ~6,300 individuals (La Marta 2; Table 1), could be considered a fourth sub-population of P. mexicana, although this hypothesis remains to be confirmed by genetic analysis.

Population size assessment of P. martinezii (e.g., populations of La Encantada and Agua de Alardín) and P. mexicana (e.g., populations of La Marta 3 and La Marta) (Table 1) support the findings of Murray & Lepschi (2004), who reported that rare species could be sparse or abundant in different locations. These dissimilar population sizes and their area extents, allowed us to identify the stands which are more prone to local extinction, given their reduced number of individuals (Table 1). Overall, the total number of individuals of P. martinezii and P. mexicana, the new reported stands and the total area occupied by both species confirm their status as rare species (Table 1). According to McCune (2016), both species could be classified as extremely rare (i.e. with less than five known populations); according to Rabinowitz (1981) both correspond to rare species that are locally abundant in specific habitats but restricted geographically, or sparse and geographically restricted in specific habitats.

Species niche requirements, distribution modelling and model assessment

The results of predictors selection by the VIM function highlight the importance of not only taking into account climatic variables to construct the distribution models for P. martinezii and P. mexicana (Table 2), thus adding to the previous knowledge of the main factors underlying the distribution of the same species (Ledig et al., 2010). According to Sexton et al. (2009), species distributions depend on many environmental factors, and it is known that while some environmental variables represent large scale processes (macro-environment), others define the micro-environmental conditions (Franklin, 2009). In the present study, models constructed for México showed that both micro- and macro-environmental variables may be the factors with the greatest influence on species distributions. For P. martinezii the most important predictor was the profile curvature (Table 2), which is a proxy of the microenvironment related to groundwater availability. For P. mexicana the main factor was minimum temperature of coldest month (Table 2), a macro-environmental variable indicating winter severity, and probably related to its adaptation to cold subalpine zones. Similar results to those observed for P. martinezii were reported for the rare species Hesperocyparis forbesii (Jeps.) Bartel (Tecate cypress), where two proxies of the micro-environment (a topographic and a soil variable) were the most important predictors of the species distribution (Regan et al., 2012). On the other hand, the results obtained for P. mexicana were similar to those reported for other widespread forest species, for which similar sets of environmental variables were used and the main drivers of species distributions were found to be climatic variables (Penteriani et al., 2019). However, there were soil and topographic variables among the most important predictors for P. mexicana.

Regarding the model metrics, some authors have prevented that the AUC tends to increase when the calibration areas are larger and further from presence records (Lobo, Jiménez-Valverde & Real, 2008; Hijmans, 2012), making preferable to consider additional model evaluators like sensitivity, specificity (Lobo, Jiménez-Valverde & Real, 2008), or TSS which accounts for both omission and commission errors and is not influenced by the sample size of each class (Allouche, Tsoar & Kadmon, 2006; Wehenkel et al., 2020). The high values of the seven metrics used to evaluate our regional and global models (including AUC, sensitivity, specificity and TSS; Table S3), suggested acceptable model performances. Lower model performances have been reported for different widespread species and similar sets of environmental variables and model metrics (Penteriani et al., 2019; López-Sánchez et al., 2021), or only climate variables and AUC (Dyderski et al., 2018). Nevertheless, similar results (only for AUC) have been obtained for Picea chihuahuana Martínez (Pinedo-Alvarez et al., 2019), another rare Mexican spruce. The overall better model performance for rare than for widespread species, observed in the comparable metrics, can be explained in part by the reliance on the spatially restricted environmental conditions of the former (McCune, 2016).

Regarding the worldwide projections, to our knowledge, this is the first time that a global study of the suitable climate habitat for a plant species has been carried out; but, similar AUC values have been reported by Taucare-Ríos, Bizama & Bustamante (2016) for an animal species.

Additionally to their high metrics, the regional models (developed with climate, soil and topographic variables) matched with observations in the field (Fig. 4), while global models (developed with climate variables) almost matched with observations on terrain (Figs. 6 and 7A) and identified highly suitable patches where other endemic spruces and similar tree communities thrive in different biogeographic regions (Lin et al., 2012; Bodare et al., 2013; Panthi et al., 2017) (Fig. 7).

Ledig et al. (2010) made habitat projections for the same two spruce species by using only climatic variables and a threshold of presence ≥ 0.5, and marked several sites near the natural populations with high potential to hold P. martinezii (e.g. a large area at the northwest of El Butano population, at the Coahuila-Nuevo León states border, México) and P. mexicana (e.g. mountains Cerro San Rafael, Cerro Potrero de Ábrego and Cerro El Potosí, all above 3,000 m of elevation, in Coahuila and Nuevo León states, México), but where this species does not occur at present. Our models showed that no habitat was available in any site near the current populations of P. martinezii using a PoPthreshold ≥ 0.5 (Fig. 4A), as confirmed by field observations. For P. mexicana, our models coincide with the projections of Ledig et al. (2010), for the sites mentioned before, at a PoPthreshold ≥ 0.5; however, the absences of P. mexicana on those mountains confirmed by field surveys were correctly predicted by our models at a PoPthreshold ≥ 0.8, which indicate that the more accurate probability of presence of P. mexicana is above this threshold.

Mapping suitable habitat: past and contemporary distributions

The reduced and scattered distributions of the highly suitable habitat for both species during the Last Glacial Maximum, the Middle Holocene and at present (Figs. 3 and 4), indicate that these species have had restricted habitat for a long period of time, and have survived in climatic refugia. These refugia appear to have been more widespread in the past for P. mexicana than for P. martinezii. Apparently, the main refugia in the past for P. mexicana was at south and central México (Fig. 3C). From that high elevations and rugged regions, likely P. mexicana migrated to its current locations. That is consistent with pollen records which suggest that members of the Picea genus reached the north of Chiapas and the south of Veracruz in the Miocene in southern México (Graham, 1976; Rzedowski, 2006) and the Texcoco lake in central México (bordering with what is today México city) in the Last Glacial Maximum (Lozano-García et al., 1993); although it would be discarded that such Picea pollen might belong to P. martinezii because of the very scattered marginal areas in the LGM and MH (Fig. 3A and 3B), it cannot be discarded for P. chihuahuana (Pinedo-Alvarez et al., 2019).

Potential for future assisted migration: projections for México and worldwide

Future projections of suitable habitat show an extremely grim situation for both species inside México; which could imply the loss of all natural populations (including recruitment, saplings and trees). For P. martinezii, there will be available only a few square kilometers of marginal habitat, and none of them near the current populations when considering climate, soil, geologic and topographic variables for modeling (Fig. 5B); however, more areas holding the suitable climatic habitat (considering only climate) will be available near the populations of Agua de Alardín, Agua Fría and La Encantada (Fig. 6, c). The reason why more areas appear available inside México, all close to the contemporary populations, is because the global modeling is more inclusive, by considering only climatic variables and excluding the proxies of micro-environment.

As for P. mexicana, since it is anticipated a complete loss of the highly suitable habitat in México by 2050 (both considering climate, soil, geology and topography, or only climate), it is needed to take into account the possibility of assisted migration outside the Mexican borders, either to the zones holding the marginal (p = 0.5–0.79) or the highly suitable habitat (p ≥ 0.8) (Fig. 7). However, the two zones holding the highly suitable habitat, i.e., Chungyang mountains in Taiwan and the Indian Himalayan region, are inhabited by other two endemic spruce species: Picea morrisonicola (Bodare et al., 2013) and Picea smithiana (Panthi et al., 2017), respectively (Fig. 7B, g, i).

Based on this, assisted migration would be an option for P. martinezii only on marginal sites in México (p = 0.5–0.79; Fig. 5B, a, b, c; and Fig. 6, c). For P. mexicana, a possibility would be continental and transcontinental translocations to sites holding the marginal or the highly suitable habitat (Fig. 7B). This rises two conservation issues: the first is related to whether or not perform assisted migration to marginal sites which do not cover the main habitat requirements for the species; the second is related to which species (the local or the foreign) should be prioritized for conservation when suitable habitat is found elsewhere but is inhabited by similar endemic species.

We argue that priority for assisted migration should be given to areas with the least potential negative impact on other local spruce species or their close relatives. Such negative implications could be: (i) the unintended introduction of pests and diseases (Simler et al., 2019) or, (ii) introgression through unintended pollen dispersal (Gómez et al., 2015). Therefore, two essential issues should be considered in potential assisted migration projects: (i) a detailed examination of the targeted suitable areas, as well as analyzing the potential impact of physiography and local-scale edaphic variables, as suggested by Wang et al. (2019) for Spiranthes parksii Correl and by Wilson, Roberts & Reid (2011) for Margaritifera margaritifera (L.); and (ii) the costs and benefits balance of this kind of projects, such as the potential negative effects that introduced species could have over local species.

This results give insights of a future biodiversity scenario, where REPS will rely on translocations beyond their native ranges (including across country borders) to subsist in nature, considering the global amount of rare species and their vulnerabilities (Işik, 2011; Enquist et al., 2019). Therefore, new mechanisms of international cooperation need to be discussed to deal with this expected crisis triggered by climatic change (Román-Palacios & Wiens, 2020; Brodie et al., 2021; McDonald & McCormack, 2021).

In this sense, international bodies such as the International Union for Conservation of Nature, the Convention on Biological Diversity, the Ramsar Convention on Wetlands, the Man and Biosphere Programme, or the United Nations Convention to Combat Desertification, would be valuable institutions in providing recommendations, guidelines, and mechanisms to reduce the loss of biodiversity, weighting the risks involved in potential assisted migration projects across country borders (Schwartz et al., 2012; Brodie et al., 2021). In general, such instruments of conservation should be based on the ecological, ethical and social implications and the cost-benefit balance that species translocations beyond nations’ borders would imply (IUCN/SSC, 2013), considering the scientific, government and stakeholders support as well as local people acceptance (Pérez et al., 2012).

Finally, a parallel strategy of in situ conservation should not be discarded as recommended by the Global Strategy for Plant Conservation (available at URL: https://www.cbd.int/gspc/), considering the possibilities for permanence in local microrefugia (Dawson et al., 2011), as reported for other rare plant species (Patsiou et al., 2014). Such in situ conservation activities could include the following: (i) protection of natural recruitment against livestock, plagues, illegal logging, and wildfire, (ii) establishment of artificial recruitment with autochthonous genotypes in well-selected sites near (but not within) the respective population, (iii) removal of competing vegetation (including other tree species) in the vicinity of the natural stands, (iv) assisting biotic dispersal vectors, and (v) monitoring the existing in situ populations (Wehenkel & Sáenz-Romero, 2012).

Potential limitations of the study

Our distribution models included only a set of climate, soil, geologic and topographic environmental factors, with more factors in models for México (42 variables) than for worldwide models (19 climate variables) (Table S4). By sampling only abiotic factors, we were not able to search for future suitable areas based on biotic factors like the amount of genetic differentiation among populations, that result in differential degrees of local adaptation (Benito-Garzón, Robson & Hampe, 2019; Zhao et al., 2020) or biotic interactions (Godsoe et al., 2017; Flores-Tolentino et al., 2020), due to the lack of such data for these species.

On the other hand, some new stands of the studied spruces occupy areas as small as 0.1 ha (Table 1), similar to microrefugia. As detection of such very small suitable areas depends on grid resolution (Franklin et al., 2013; Patsiou et al., 2014), our models could not identify these microhabitats outside the species ranges, with a resolution of 30 arc-seconds (≈0.7 km2 or 70 ha). Moreover, SDMs are highly susceptible to produce different results in their geographic projections, and future suitable areas for the species will depend on the used method (Pearson et al., 2006; Qiao, Soberón & Peterson, 2015). Hence, we cannot exclude that there are or will be even better models. However, the findings of potential areas for P. mexicana in zones with very similar spruce forest communities in Asia is a good indicator of the quality of these models.

Conclusions

Our findings confirm that Picea martinezii and Picea mexicana are narrow endemics with varying populations sizes, but viable total populations (Table 1), and suggest that the habitats of both tree species were limited since the last glacial age (Figs. 2 and 3). Considering the surface areas holding the highly suitable habitats (p ≥ 0.8), contemporary conditions appear to be more suitable than conditions during the Last Glacial Maximum and Middle Holocene for P. martinezii; and vice versa for P. mexicana (Fig. 2). Current highly suitable areas will mostly disappear in the near future (Figs. 2 and 5). For Picea martinezii, the possibility for future assisted migration in northern and central México is only on marginal sites (p = 0.5–0.79), all far from its current distribution. Regarding Picea mexicana, a complete disappearance of the suitable habitat within México is anticipated; hence, it is needed to discuss the possibility of species translocations beyond the national borders of México, to sites holding the intermediated (p = 0.5–0.79) or the highly (p ≥ 0.8) suitable climatic habitat, even considering sites so far away as The Himalayans or Taiwan. In the expected stage where species translocations will become necessary to avoid the high extinction rates because of climate change, new mechanisms of international cooperation need to be discussed. In this sense, institutions similar to the International Union for Conservation of Nature, the Convention on Biological Diversity, the Ramsar Convention on Wetlands, the Man and Biosphere Programme, or the United Nations Convention to Combat Desertification would promote this international collaboration and set guidelines and recommendations in assisted migration projects. Meanwhile, in situ conservation should not be discarded, considering marginal microhabitat sites. Future decisions for ex situ conservation would be reinforced with data from common garden assays displaying the species’ resilience to environmental gradients.

Supplemental Information

Supplemental Information 1 Distribution records for Picea martinezii obtained from field surveys.

Click here for additional data file.

Supplemental Information 2 Distribution records for Picea mexicana obtained from field surveys.

Click here for additional data file.

Supplemental Information 3 Model fit metrics for species distributions as indicated by Random Forest analysis applied to the occurrence data for Picea martinezii and Picea mexicana, with a cross validation n = 10.

Model fit metrics included the area under the receiver operator curve (AUC), the overall accuracy (OA), Matthews correlation coefficient (MCC), true skill statistic (TSS), Cohen’s kappa, sensitivity, specificity and probability of presence (PoP).

Click here for additional data file.

Supplemental Information 4 Descriptive statistics of the 42 environmental variables used to characterize the abiotic niches and to construct the distribution models of Picea martinezii and Picea mexicana, from the presence locations. Min = minimum, SD = standard deviati.

Click here for additional data file.

Supplemental Information 5 Model projections for the contemporary conditions in México (using climate, soil, geological and topographic variables).

The categories of probability of presences (p), the pixel count for each category of presence, and the real presences/absences of Picea martinezii and Picea mexicana on the corresponding predicted areas by the models are shown.

Click here for additional data file.

Additional Information and Declarations

Competing Interests

Author Contributions

Data Availability

Christian Wehenkel is an Academic Editor for PeerJ.

Eduardo Mendoza-Maya conceived and designed the experiments, performed the experiments, analyzed the data, prepared figures and/or tables, authored or reviewed drafts of the article, and approved the final draft.

Erika Gómez-Pineda analyzed the data, prepared figures and/or tables, authored or reviewed drafts of the article, and approved the final draft.

Cuauhtémoc Sáenz-Romero analyzed the data, authored or reviewed drafts of the article, and approved the final draft.

José Ciro Hernández-Díaz analyzed the data, authored or reviewed drafts of the article, and approved the final draft.

Carlos A. López-Sánchez performed the experiments, analyzed the data, prepared figures and/or tables, authored or reviewed drafts of the article, and approved the final draft.

J. Jesús Vargas-Hernández analyzed the data, authored or reviewed drafts of the article, and approved the final draft.

José Ángel Prieto-Ruíz analyzed the data, authored or reviewed drafts of the article, and approved the final draft.

Christian Wehenkel conceived and designed the experiments, performed the experiments, analyzed the data, prepared figures and/or tables, authored or reviewed drafts of the article, and approved the final draft.

The following information was supplied regarding data availability:

The raw data to construct the distribution models are available in the Tables S1 and S2 (presences records); the repository of the absences records are available in the Materials & Methods.

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
