# Peer review of "Assisted migration and the rare endemic plant species: the case of two endangered Mexican spruces"

_PeerJ, doi:10.7717/peerj.13812_

## Round 0.1 · original submission · Major Revisions

· Academic Editor

Major Revisions

Dear Authors

I have received two reviews on your manuscript: both reviewers point out that it is interesting research although they come to an overall opposite assessment. In particular, the main flaws focus on the method section. I agree that the manuscript is interesting and presents a very stimulating topic in conservation of threatened plants but I believe that reviewer 2 has correctly focused on numerous shortcomings in the methods. For this reason, I want to offer the authors a chance to present a new version of this manuscript, working deeply on the method section, starting from the suggestions of reviewer 2.

In addition, I highlight some additional elements that need revision:

- considering that the research presents a single count of individuals (and not on the analysis of population dynamics) it is more appropriate to speak of "population size assessment";

- the population assessment only covered saplings and trees while I did not find any reference to natural recruitment; have you observed recruitment? if so, what impact could this factor have in future projections?

- a section / paragraph explaining the limitations of the model is required.

- LL104-107: this sentence needs a methodological reference; the cited paper (Fois et al., 2018) simply reports this sentence and must be deleted and replaced with a consistent reference. Similarly, the same reference is given for assisted migration (LL113-116) but this review does not deal with this issue. There is an extensive bibliography on this subject.

Reviewer 1 has suggested that you cite specific references. You are welcome to add it/them if you believe they are relevant. However, you are not required to include these citations, and if you do not include them, this will not influence my decision.

Reviewer 1 ·

Basic reporting

no comment

Experimental design

no comment

Validity of the findings

no comment

Additional comments

The manuscript “ Assisted migration and the rare endemic plant species: The case of two endangered Mexican spruces” is well organized and well written.
The authors have used demographic censuses, field surveys in the search for new stands, and
developed distribution models to achieve the following aims: 1) (i) the exploration of potential new stands and performance of complete demographic censuses to determine individuals numbers, size, and extent of all populations; (ii) construction of models describing the past and current distribution of these species to gain some insight about their distribution dynamics over time; and (iii) identification of areas outside their current ranges (inside Mexico and worldwide) with a high probability of harboring suitable habitats for future assisted migration, by modelling future distributions.
I have some specific comments on this manuscript. Please, see below.
Keywords: they are not keywords that highlight the work
Line 125: What were the factors that explain the extinction? Can you identify them?
Lines 480-496 : this part is a bit confusing, please reduce and make it clearer
Line 470: Our position is that those zones.. Please motivate more strongly
497-499: You affirm that “finally, a parallel strategy of in situ conservation should not be discarded" and In the Conclusion section affirm that " Meanwhile, in situ conservation should be prioritized (Line 519). Although you highlight the point as fundamental in the conclusions, neither you do not emphasize it enough either in the introductory part or in the discussions. Considering that according to Article 8 of the GSPC, in situ conservation is generally considered the primary approach for species conservation as it ensures that species are maintained in their natural environments while ex situ conservation plays a complementary role in situ conservation, providing a “safety back -up ”and an insurance policy against extinction in the wild, can you better rephrase this whole part?
Line 501: You need a stronger conclusion here

Please, add some other references to support the MS:
DOI : https://doi.org/10.1556/168.2018.19.3.3
DOI : doi.org/10.1007/s11258-016-0589-6.
DOI 10.1007/s10531-019-01757-0
DOI : http://dx.doi.org/10.1016/j.foreco.2017.06.044.

Reviewer 2 ·

Basic reporting

Abstract.
The beginning of this phrase is confusing: In the coming climatic scenarios, assisted migration would play an important role in the conservation of the threatened plant species… What do you mean by “coming scenarios''? Are you referring to the CMIP6? I saw you used CMIP5. Or do you refer to global warming?

In the background, you also mention “... distribution dynamics”, but I didn’t find any related to this.

Introduction
This section is well organized, but there are points that need to be strengthened and complemented, see above.

Please include in the introduction the non-positive implications of Rescue assisted migration, outside species range, and the country to which it is endemic.
SDM are highly susceptible to producing different results in their geographic projections, future suitable areas for the species will depend on the used method (see https://onlinelibrary.wiley.com/doi/abs/10.1111/j.1365-2699.2006.01460.x). This is a crucial point in your assessment that you didn't consider thus it reduces its robustness.

Line 133-135. Because you are modeling a only climate scenario, habitat is not a word that comprehensively describes the geographic hypothesis (SDM)

Experimental design

Methods

There are several sections of the methods that are missing or ambiguous.
Lines 153-158. For clarity, please include here how many presence and absence records were used to calibrate RF. How did you consider spatial autocorrelation of species occurrence data? Additional, for RF, Barbet-Massin https://besjournals.onlinelibrary.wiley.com/doi/full/10.1111/j.2041-210X.2011.00172.x recommend: “using the same number of pseudo-absences (or absences) as available presences (averaging several runs if few pseudo-absences) for classification techniques such as boosted regression trees, classification trees and random forest”, but as I mentioned before I do not understand how many presence were used.

Line 185. Specify did you aggregate the cell grid to match 30 arc sec. What did you use, mean, median...?
Lines 196-200. I understand that with this method you select a subset of variables from the initial 42 environmental variables. Please specify which were variables used for each species, in the regional and global analysis. Also, it seems that you didn't reduce collinearity between variables, which is an important part of the modeling process mainly if the aim is to understand which variable limits species distribution.
Lines 200-203. This part is confusing. As I understand it VIM values are just a measure of the contribution of the variables to the model, please correct me if I am wrong. So I don't get how it is possible to normalize VIM values. Or did you normalize the environmental variables?
Lines 205-220. Everything seems OK here, but till the number of unique (unique in a grid-cell of 1 km2 resolution) occurrences (presence and absences) are clear it is very hard to be sure. For the number of stands it seems you used very few records.

Section 2.3

This section needs to be improved. Past and future GCM represents very different possible climate conditions, thus they are a high source of uncertainty. There have been several recommendations to incorporate or show climate storytellings, https://besjournals.onlinelibrary.wiley.com/doi/abs/10.1111/2041-210X.13360. It is not clear which and how many GCM you used.

Validity of the findings

Please considered all the observations related to your methods, this must be acknowledged to improve the validity of your findings

Additional comments

none

---

## Round 0.2 · Minor Revisions

· Academic Editor

Minor Revisions

Dear Authors,

Both reviewers now agree that the manuscript may be accepted for publication. However, I kindly ask you for a further effort to address the last point, highlighted by reviewer 2, which seems to me important in the context of the manuscript. Please discuss the issue of high AUC values as a consequence of the calibration area in a few sentences. Some relevant rarticles that can guide your work, have been suggested by Reviewer 2.

I look forward to the new version of the manuscript.

Reviewer 1 ·

Basic reporting

This manuscript now is clearer. The overall idea is very interesting.

Experimental design

The methodology is well described.

Validity of the findings

The manuscript, in general, is clear and well-structured as also the interpretation and the discussion of the results.

Reviewer 2 ·

Basic reporting

All comments by the reviewers have been answered appropriately. Results are nicely discussed by the authors, please only adrees my additional comment.

Experimental design

I have nothing to add to this version.

Validity of the findings

Please see additional comments.

Additional comments

You give too much weight to the performance of your model in the discussion, however, you miss to discuss that the high values of for example the AUC are a product of the calibration area. High AUC values can be obtained when calibration áreas are larger and further from presence records.
See https://esajournals.onlinelibrary.wiley.com/doi/10.1890/11-0826.1 and Jorge Lobo's et al. famous paper https://onlinelibrary.wiley.com/doi/abs/10.1111/j.1466-8238.2007.00358.x?casa_token=edB-w_hRSJQAAAAA:_T8sGSISwou-ZrcZBnuDfKDmQFl6aEIQMe9wGy2p4NlD74xgK5ZDnbXC8HKQdEpe5xrYuvVT4lQPMxLo

---

## Round 0.3 · accepted · Accept

· Academic Editor

Accept

Dear Authors,

Thanks for the work done, I believe that the new version of the manuscript is ready for acceptance.

Best Regards